# LANGUAGE MODELS CAN EXPLAIN VISUAL FEATURES VIA CAUSAL INTERVENTIONS

## ABSTRACT

Sparse Autoencoders uncover thousands of features in vision models, yet explaining these features without requiring human intervention remains an open challenge. While previous work has proposed generating correlation-based explanations based on top activating input examples, we present a fundamentally different alternative based on causal interventions. We leverage the structure of Vision-Language Models and *steer* individual SAE features in the vision encoder after providing an empty image. Then, we prompt the language model to explain what it "sees", effectively eliciting the visual concept represented by each feature. Results show that *Steering* offers an scalable alternative that complements traditional approaches based on input examples, serving as a new axis for automated interpretability in vision models. Moreover, the quality of explanations improves consistently with the scale of the language model, highlighting our method as a promising direction for future research. Finally, we propose *Steering-informed Top-k*, a hybrid approach that combines the strengths of causal interventions and input-based approaches to achieve state-of-the-art explanation quality without additional computational cost.

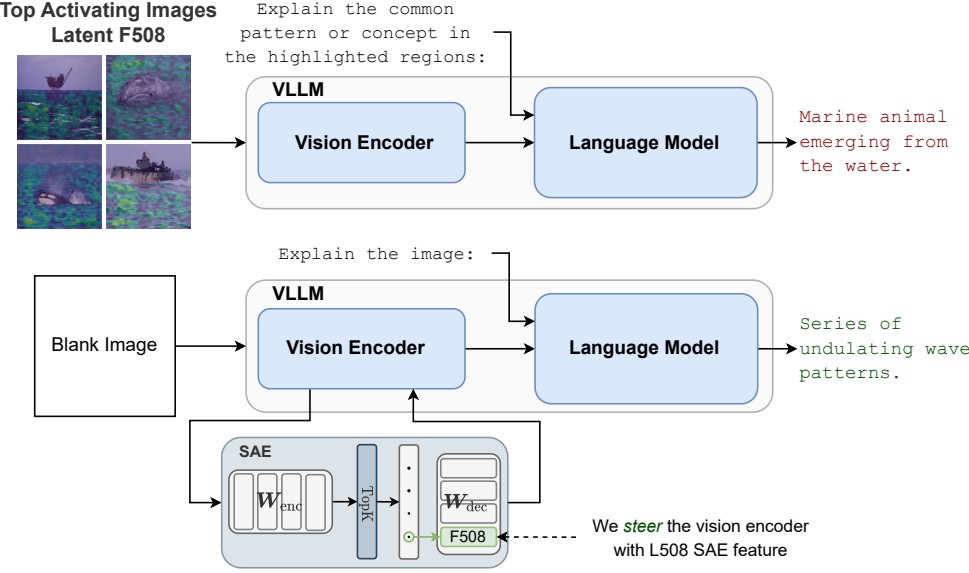

Figure 1: We propose to automatically obtain explanations of SAE features by causally intervening (*steering*) a vision encoder. The intervention is done after feeding it an information-devoid 'blank image', effectively making the language model articulate what visual concept that feature represents.

## 1 INTRODUCTION

Understanding what features neural networks learn is a central goal in interpretability research (Olah et al., 2020). Sparse Autoencoders (SAEs) have emerged as a promising unsupervised method for uncovering human-interpretable features from model representations (Bricken et al., 2023; Huben

et al., 2024), particularly in large language models (LLMs). SAEs have recently been extended to vision models, revealing semantically meaningful concepts such as object categories, patterns, and textures (Fry, 2024; Lim et al., 2025). However, as SAEs scale to uncover thousands of features, interpreting these poses significant challenges, necessitating the development of additional tools.

Recent work on automated interpretability aims to address this challenge by leveraging powerful language models as *explainers* to generate descriptions of features learned by a *subject* model (Bills et al., 2023; Paulo et al., 2024). When the subject is a vision model, the images that activate most each feature are analyzed by an explainer which looks for common patterns that may explain the target feature (Xu et al., 2025; Zhang et al., 2024). This input-based strategy relies heavily on a predefined test set, is fundamentally correlation-based rather than causally grounded, and incurs significant computational cost.

In this paper, we propose a new approach for *self-explaining vision features*[1]. Instead of interpreting features via their top activating images, we leverage the structure of VLMs and causal interventions to directly generate natural language explanations. By *steering* the vision encoder's residual stream with individual SAE features —while feeding it an empty image— we prompt the VLM to describe what visual concept that feature represents (Figure 1). Experiments on Gemma 3 and Intern VL3 vision encoders show that *Steering* offers an scalable alternative that complements traditional approaches based on input examples, overcoming some of the explanation biases these methods introduce, while surfacing lower-level features. Furthermore, scaling the language model consistently improves explanation quality, highlighting this causal, output-centric approach as a promising direction for automated interpretability.

Building on this idea, we also introduce a hybrid strategy —*Steering-informed Top-k* — that combines the best of both approaches. We condition the VLM on the top activating images *and* the causal intervention with the SAE feature, improving the quality of the generated explanations on four complementary metrics.[2]

## 2 EXTRACTING FEATURES

Model neurons often exhibit polysemanticity, meaning they respond to seemingly unrelated concepts. One leading explanation for this phenomenon is *superposition*, the idea that models learn to represent more concepts than they have neurons (Arora et al., 2018; Elhage et al., 2022). Sparse Autoencoders (SAEs) (Bricken et al., 2023; Huben et al., 2024) have emerged as an interpretability tool for finding interpretable and monosemantic features that are otherwise represented in superposition. SAEs achieve this by mapping model representations $\boldsymbol{z} \in \mathbb{R}^d$ into a higher-dimensional latent space $\mathbb{R}^{d_{SAE}}$, while enforcing sparsity in the latent representation. In this work, we use TopK SAEs (Gao et al., 2025), which apply the TopK activation function to enforce sparsity. The encoder first computes a sparse code using:

$$f(\boldsymbol{z}) = \text{TopK}\big(\boldsymbol{z}\boldsymbol{W}_{\text{enc}} + \boldsymbol{b}_{\text{enc}}\big),\tag{1}$$

and the decoder reconstructs the original input from the sparse representation via

$$\text{SAE}(\boldsymbol{z}) = f(\boldsymbol{z})\boldsymbol{W}_{\text{dec}} + \boldsymbol{b}_{\text{dec}}.\tag{2}$$

The encoder and decoder are parameterized by weight matrices and bias vectors $\boldsymbol{W}_{\text{enc}}, \boldsymbol{b}_{\text{enc}}$ and $\boldsymbol{W}_{\text{dec}}$, $\boldsymbol{b}_{\text{dec}}$ respectively. We refer to *SAE feature activation* to a component in $f(\boldsymbol{z}) \in \mathbb{R}^{\text{SAE}}$, while a *SAE feature* denotes a row vector in the dictionary $\boldsymbol{W}_{\text{dec}} \in \mathbb{R}^{d_{\text{SAE}} \times d_{\text{model}}}$. In this work, we train TopK SAEs with the latent space dimensionality $d_{\text{SAE}} = 8,192$ on Imagenet dataset (Deng et al., 2009). We refer to Section A for further training details.

## 3 AUTOMATICALLY INTERPRETING FEATURES

Following previous work in automated interpretability, we assume features can be explained by a sequence of words **e**. We consider a *subject* model $m_{\text{subj}}$ whose features we want to interpret, and an *explainer* model $m_{\text{exp}}$ that generates the natural language explanations for these features.

---

[1]We refer to a 'feature' as a direction in the model's representation space.

[2]Our code will be released upon publication.

### 3.1 *Top-k* EXPLANATIONS

The existing approach to generate explanations from vision model features (Zhang et al., 2024; Xu et al., 2025) assumes access to an evaluation set of images, $\mathcal{D}^{\text{eval}}$. Each image $I \in \mathcal{D}^{\text{eval}}$ is fed into the subject model $m_{\text{sub}}$, and the representations from the residual stream at a particular layer $l$ and position $j$, $m_{\text{sub}}^{l,j}(\cdot)$ are extracted; for brevity, we omit the layer index in what follows. Following Equation (1), a SAE feature activation vector is obtained for each position $j$, $f(m_{\text{sub}}^{j}(I)) \in \mathbb{R}^{\text{SAE}}$. For each dimension $i \in \{1, \ldots, d_{\text{SAE}}\}$, we compute an *image activation score* by aggregating the individual position activations across the entire image:

$$S^{i,I} = g\left(f_i(m_{\text{sub}}(I))\right). \tag{3}$$

Typically, the mean function (across positions) is used as $g(\cdot)$ (Zhang et al., 2024). Then, we identify the top-$k$ images (with $1 \leq k \leq |\mathcal{D}^{\text{eval}}|$) that produce the highest *image activation scores*. These images, denoted $\mathcal{T}_i^k = \{I_1^i, \ldots, I_k^i\}$, are selected such that their scores follow the descending order: $S^{i,I_1^i} \geq S^{i,I_2^i} \geq \cdots \geq S^{i,I_{|\mathcal{D}^{\text{eval}}|}^i}$. A natural language explanation $\mathbf{e}_i$ for the $i$-th feature is then generated by conditioning the explainer model on both a prompt $P$ and the selected top-$k$ images:

$$\mathbf{e}_i \sim m_{\text{exp}}(\mathbf{e} \mid P, \mathcal{T}_i^k). \tag{4}$$

Alternatively, the top-$k$ images can be modified to emphasize the regions where the feature is active. In our experiments, we explore two of such variants: 'Masks', where all non-activating patches are occluded; and 'Heatmaps', where activation intensity is overlaid to highlight the most responsive regions (see top activating images in Figure 1).

### 3.2 PROPOSED APPROACH

Current VLMs align a visual encoder with a pre-trained language model backbone (Bai et al., 2025; Team et al., 2025), enabling natural image interpretation. We hypothesize that the language model can serve as an explainer for SAE features. We do so by causally intervening the vision encoder's forward pass with each feature. We introduce two complementary methods for doing so.

***Steering*-based Explanations.** In the basic setting, we prompt[3] the model to explain an empty image $\tilde{I}$, and intervene the forward pass by adding the SAE feature vector $\boldsymbol{W}_{\text{dec}}[i, :]$ across all positions, effectively generating an explanation of the intervened feature. The process is formalized as follows:

$$\mathbf{e}_i \sim m_{\text{exp}}\left(\mathbf{e} \mid \overbrace{P}^{\text{Prompt}}, \overbrace{\tilde{I}}^{\text{Empty image}}, \overbrace{\text{do}(m_{\text{sub}}^l(\tilde{I}) = m_{\text{sub}}^l(\tilde{I}) + \alpha \boldsymbol{W}_{\text{dec}}[i, :])}^{\text{Causal intervention with SAE feature}}\right), \tag{5}$$

where we express the intervention using the do-operator (Pearl, 2009), and $\alpha$ is a coefficient indicating the strength of the intervention.[4] This method offers an efficient and scalable means of obtaining feature explanations, requiring a single forward-pass (see Appendix F for details). Unlike prior methods, it doesn't require an evaluation image set, simplifying the interpretability pipeline.

***Steering-informed Top-k* Explanations.** Instead of only using a blank image, we apply the same causal intervention while conditioning on the top-$k$ images, $\mathcal{T}_i^k$—those that most strongly activate the $i$-th SAE feature. Intuitively, this focuses the explainer on the salient concept captured by the feature, enabling more targeted and meaningful interpretations. The process is defined as:

$$\mathbf{e}_i \sim m_{\text{exp}}\left(\mathbf{e} \mid \overbrace{P}^{\text{Prompt}}, \overbrace{\mathcal{T}_i^k}^{\text{Top-k images}}, \underbrace{\text{do}(m_{\text{sub}}^l(\mathcal{T}_i^k) = m_{\text{sub}}^l(\mathcal{T}_i^k) + \alpha \boldsymbol{W}_{\text{dec}}[i, :])}_{\text{Causal intervention with SAE feature}}\right). \tag{6}$$

---

[3] The prompts used for each method can be found in Section G.

[4] In practice, we select the $\alpha$ coefficient on a validation set of 500 features.

## 4 EVALUATING THE QUALITY OF THE EXPLANATIONS

### 4.1 EVALUATION METRICS

To quantitatively assess the explanations, we adopt three complementary evaluation techniques. The first two are existing input-based evaluations relying on top-$k$ images (Zhang et al., 2024; Xu et al., 2025). To avoid evaluating on the same set of images used for extracting the explanations, we use the 50k-image Imagenet test set, $\mathcal{D}^{\text{test}}$. Finally, building on top of recent work (Shaham et al., 2024; Bai et al., 2024), we propose a pair of metrics based on synthetic images generated by diffusion model.

**Simulation-based Evaluation.** Zhang et al. (2024); Xu et al. (2025) propose using a segmentation model $m_{\text{seg}}$, (e.g., SAM 2 (Ravi et al., 2025a)) to generate binary masks $M_{\text{seg}}$ containing 1s on the image patches that correspond to the concepts described in the explanations. These masks simulate how the SAE feature would activate if the explanation were true. $M_{\text{seg}}$ masks are compared against the actual feature's activation masks $M_{\text{feature}}$. More formally, given an image and an explanation, the masks are computed as follows:

$$M_{\text{feature}}^{i,I} = \mathbb{1}[f_i(m_{\text{sub}}(I)) > 0], \quad M_{\text{seg}}^{i,I} = m_{\text{seg}}(I, \mathbf{e}_i), \tag{7}$$

where $\mathbb{1}[\cdot]$ is an indicator function that returns 1 if the condition holds and 0 otherwise. To quantitatively assess the alignment between these simulated and actual activation masks, the Intersection over Union (IoU) is computed and averaged over the top-k activating images $\mathcal{T}_i^k$ on $\mathcal{D}^{\text{test}}$:

$$\text{IoU-Score}^i = \frac{1}{k} \sum_{I \in \mathcal{T}_i^k} \frac{|M_{\text{seg}}^{i,I} \cap M_{\text{feature}}^{i,I}|}{|M_{\text{seg}}^{i,I} \cup M_{\text{feature}}^{i,I}|}. \tag{8}$$

**CLIP-based Evaluation.** To assess the semantic alignment between explanations and the corresponding top-$k$ activating images in $\mathcal{D}^{\text{test}}$, we follow Zhang et al. (2024) and use a CLIP model $m_{\text{clip}}$. For each dimension $i$, we compute the text embedding from the explanation $\mathbf{e}_i$ and extract visual embeddings from the top-$k$ activating images $\mathcal{T}_i^k$ associated with that feature. Specifically, for each image $I \in \mathcal{T}_i^k$, we apply the feature's activation masks ($M_{\text{feature}}^{i,I}$) to focus on the relevant region, and compute its CLIP image embedding. We then measure the cosine similarity between the explanation embedding and each masked image embedding, averaged across images:

$$\text{CLIP-Score}^i = \frac{1}{k} \sum_{I \in \mathcal{T}_i^k} \cos\left(m_{\text{clip}}^{\text{text}}(\mathbf{e}_i), m_{\text{clip}}^{\text{img}}(I)\right). \tag{9}$$

**Synthetic-image-based Evaluation.** For each feature $i$, we generate a set of $N$ *positive* images using a diffusion model[5] $m_{\text{diff}}$ conditioned on the explanation $\mathbf{e}_i$, $\mathcal{I}^{i,+} = \{I \sim m_{\text{diff}}(I \mid \mathbf{e}_i)\}^N$. Then, we compute the average feature (synthetic) image activation score (Equation (3)):

$$\text{Synthetic-Activation-Score}^i = \frac{1}{N} \sum_{I \in \mathcal{I}^{i,+}} S^{I,i}. \tag{10}$$

We also generate a set of $N$ *negative* images, $\mathcal{I}^{i,-} = \{I \sim \mathcal{D}^{\text{test}}\}^N$ by randomly sampling from the test set. Following Equation (3), we obtain the *image activation score* for each positive and negative image and repeat the process for every feature. Finally, we compute the AUROC metric.[6]

### 4.2 EXPERIMENTAL SETUP

We train SAEs on a middle-layer of the vision encoders of Gemma 3 (Team et al., 2024) and the InternVL3-14B (Zhu et al., 2025), two state-of-the-art VLMs. We also train a SAE at a later layer (3/4th depth) of Gemma 3 encoder. Gemma 3 employs a 400M parameters variant of the SigLIP encoder (Zhai et al., 2023), which works at a fixed resolution of $896 \times 896$ pixels. It remains

---

[5]We use Stable Diffusion 3.5 Medium (Esser et al., 2024).

[6]This is mathematically equivalent to the probability that the obtained image activation score for a 'positive' image in $\mathcal{I}^{i,+}$–generated by $m_{\text{diff}}$–ranks higher than a randomly chosen negative image from $\mathcal{D}^{\text{test}}$.

Table 1: Explanation evaluation metrics for the middle layer SAE of Gemma 3 and InternVL3-14B vision encoders. Except for AUROC, mean scores are reported, and statistical significance is assessed pairwise between methods. A value is underlined if it is significantly higher (with $p < 0.05$) than both other methods in the same column.

| Model | Explanation Method | IoU Score | | AUROC | | Synth. Act. Score | | CLIP Score | |
|---|---|---|---|---|---|---|---|---|---|
| | | Masks | Heatmaps | Masks | Heatmaps | Masks | Heatmaps | Masks | Heatmaps |
| Gemma 3 | Steering | 0.211 | | 0.675 | | 0.324 | | 0.186 | |
| | Top-k | 0.211 | 0.198 | 0.723 | 0.791 | 0.330 | 0.364 | 0.190 | 0.187 |
| | Steering-informed Top-k | 0.216 | 0.203 | 0.788 | 0.838 | 0.461 | 0.505 | 0.193 | 0.189 |
| InternVL3 | Steering | 0.220 | | 0.655 | | 0.141 | | 0.191 | |
| | Top-k | 0.224 | 0.201 | 0.768 | 0.775 | 0.187 | 0.183 | 0.199 | 0.187 |
| | Steering-informed Top-k | 0.228 | 0.203 | 0.823 | 0.833 | 0.254 | 0.252 | 0.199 | 0.191 |

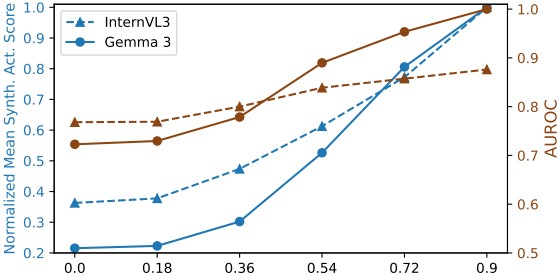

Figure 2: Middle layer SAE synthetic-image-based evaluation scores of *Top-k* method as a function of the similarity with *Steering* Explanations.

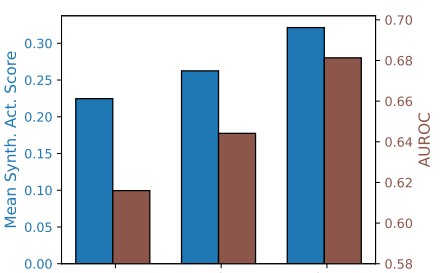

Figure 3: Gemma 3 synthetic-image-based evaluation scores of *Steering* method as a function of the size of the LM $m_{subj}$.

frozen during LM training and adaptation stages and produces 4,096 tokens per image. In contrast, InternVL3-14B incorporates the pretrained InternViT-300M-448px-V2_5 encoder (300M parameters), which processes images at a $448 \times 448$ resolution, producing 256 tokens per input. This setup enables us to evaluate our proposed methods on a 'pure' SigLIP encoder (Gemma 3) and another encoder adapted through joint training (InternVL3).

Unless stated otherwise, the explainer models correspond to the same VLM from which the encoder is interpreted, Gemma 3 27B and InternVL3-14B respectively. The prompts used for *Top-k* and *Steering-informed Top-k* (these two methods share the same prompt) are designed to closely mirror that of *Steering*, ensuring consistency across methods (see Section G). For all experiments involving top-activating images, we report results using the top five images (i.e., $k$=5). Visualizations of the explanations produced by the different methods, alongside the top-activating images, are available in the demo accompanying the paper[7] (see Appendix H for more information).

## 5 RESULTS

To compare the different explanation methods we evaluate the quality of the explanations generated by these methods using the metrics described in Section 4. Our analysis is divided in three parts. Section 5.1 evaluates the performance of the *Steering* method and illustrates its potential to reduce *contextual bias* present in standard *Top-k* explanations. Section 5.2 shifts focus to the *Steering-informed Top-k* method, showing how it improves explanation quality. Finally, Section 5.3 explores the SAE feature space to uncover the semantic structure of learned features.

---

[7]anonymized link

| Model | Masking Type | Steering | | Top-k | | Steering-informed Top-k | |
|---|---|---|---|---|---|---|---|
| | | Count | % | Count | % | Count | % |
| Gemma 3 | Masks | 0 | 0.0% | 23 | 7.7% | 12 | 4.0% |
| | Heatmaps | 0 | 0.0% | 125 | 47.7% | 116 | 38.6% |
| InternVL3 | Masks | 0 | 0.0% | 19 | 6.3% | 13 | 4.3% |
| | Heatmaps | 0 | 0.0% | 125 | 41.7% | 117 | 39.0% |

Table 2: Count and percentage of 'background' explanations turned 'animal' explanations by different methods (see main text for details).

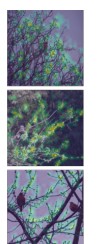

Figure 4: Example of *Top-k* explanation exhibiting *contextual bias*.

**Top-K**
Bird perched on a branch amidst blossoms.
**Steering**
Dense cluster of thin, intertwining branches with small, dark berries.
**Steering-informed Top-k**
Flowering tree branch.

## 5.1 EXPLAINING THROUGH STEERING

We analyze the effectiveness of the *Steering* method focusing on how it scales with model size, performs across different evaluation metrics, and complements *Top-k* explanations. Our results suggest that, although *Steering* has limitations when used in isolation, it scales effectively, inherently mitigates *contextual biases*, and can be used to improve other interpretability methods, despite being more efficient.

**Steering performs well on IoU but lags behind in the rest of metrics.** While *Steering* performs competitively on IoU Score, especially on later layer SAE Section C, it consistently lags behind in the remaining metrics. This is evident across all evaluated layers and models, where *Steering* achieves solid overlap with segmentation masks but fails to elicit strong activations or achieve high AUROC and CLIP alignment. In particular, its AUROC and Mean Synthetic Activation scores are substantially lower than those of *Steering-informed Top-k*, indicating weaker model sensitivity and less effective explanation quality.

**Steering helps surface high-quality Top-k explanations.** We hypothesize that the *Steering* method can act as a valuable signal for validating *Top-k* explanations. Intuitively, if both methods independently produce semantically similar explanations for the same feature, this agreement may indicate a higher likelihood of correctness. To test this, we compute semantic embeddings of each explanation using a sentence similarity model (Reimers and Gurevych, 2019) and measure the semantic similarity between the explanations produced by *Top-k* and *Steering*. We then assess the quality of the *Top-k* explanations as a function of this similarity, retaining only those above varying thresholds.

As shown in Figure 2, explanation quality—measured by normalized synthetic activation scores and AUROC—improves consistently as the similarity to *Steering* increases. This trend holds across both the Gemma 3 and InternVL3 encoders with the exception of Gemma's CLIP Score Section D. These results suggest that *Steering* serves as an effective filter or guide, helping to identify high-quality explanations and improving the overall interpretability pipeline when used in conjunction with *Top-k*.

**Steering quality scales with LM size.** *Steering* explanations improve as the size of the underlying language model used for generation increases. In this experiment, we vary the size of the LM used to produce explanations while keeping all other components fixed. As shown in Figure 3, both evaluation metrics—Mean Synthetic Activation Score and AUROC—show consistent improvements when moving from 4B to 12B to 27B parameter models. A positive trend is also observed for the rest of the metrics in Section E. This suggests that larger language models generate more informative and causally effective explanations when used in the *Steering* framework. Crucially, this trend points to a promising direction: as language models continue to grow in scale and capability, we can expect the quality of *Steering*-based interpretability to improve accordingly.

**Steering prevents contextual biases found in Top-k.** To better understand what *Steering* captures that *Top-k* does not, we analyze the 300 features with the largest IoU score difference between the two methods. Manual inspection of this subset reveals that *Steering* often produces accurate *background* explanations, whereas *Top-k* tends to misattribute these features to foreground elements such as animals, likely due to recurring context in the top activating images, a pattern we name *contextual bias* (see Figures 1 and 4).

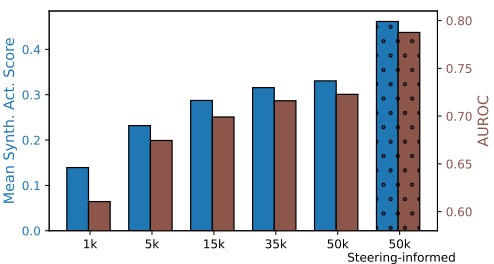 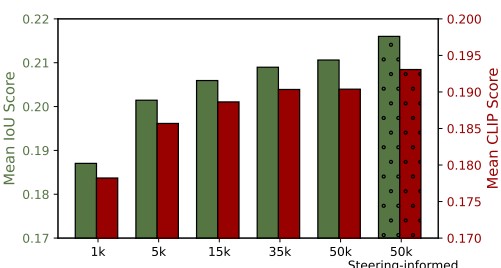

Figure 5: Explanation evaluation scores–synthetic-image-based scores on the left, and IoU and CLIP scores on the right of *Top-k* method as a function of the evaluation set size. *Steering-informed Top-k* results on the rightmost bar.

To quantify this effect, we categorize each explanation using Gemma 3 27B as *background*, *animal*, or *other*. As shown in Table 2, *Top-k* explanations frequently fall into the *animal* category (e.g., 47.7% with heatmaps), despite the feature aligning with *background* under *Steering* with a high IoU score. Notably, the hybrid *Steering-informed Top-k* reduces this misattribution (to 38.6%), suggesting it inherits some of *Steering*'s robustness to contextual bias.

## 5.2 THE BEST OF BOTH WORLDS: *Steering-informed Top-k*

We now analyze how the *Steering-informed Top-k* method consistently improves explanation quality. In this section, we highlight two key findings: the consistent superiority of *Steering-informed Top-k* across all metrics, and its ability to overcome the diminishing returns when using larger datasets.

**Steering-informed Top-k gives the best explanations across the board.** Across models and layers, *Steering-informed Top-k* consistently achieves the best performance across all evaluation metrics—IoU Score, AUROC, Mean Synthetic Activation, and CLIP Score—demonstrating its superiority in producing high-quality explanations. In both the middle and later layers of the Gemma 3 vision encoder (Table 1 top, and Table 3), as well as in the middle layer of the InternVL3-14B encoder (Table 1 bottom), this method outperforms both standard *Steering* and *Top-k* approaches, regardless of whether masks or heatmaps are used. Notably, it achieves the highest AUROC and Synthetic Activation scores, indicating that the explanations not only align well with segmentation and top-$k$ activating images, but also elicit stronger feature activations when using synthetic examples. These results underline the effectiveness of combining top-k selection with causal interventions to enhance explanation quality.

**Steering-informed Top-k overcomes diminishing returns.** We additionally generate *Top-k* and *Steering-informed Top-k* explanations using the top-k images obtained with a reduced evaluation dataset. As observed in Section 5.2, as the size of the evaluation dataset increases, standard *Top-k* explanations gradually improve in quality, but the gains exhibit diminishing returns, especially beyond 15k examples. This trend is visible across all metrics. In contrast, *Steering-informed Top-k* provides an immediate and substantial performance boost, effectively bypassing the need for large-scale data to reach high-quality explanations, with particular improvements in synthetically generated metrics (Section 5.2), suggesting that the causal intervention adds valuable signal beyond what dataset scaling alone can offer.

## 5.3 EXPLORING THE SAE FEATURE SPACE

To complement the previous evaluations, this section provides an overview of the structure of the learned SAE feature space. For this purpose, we use the middle-layer SAE of Gemma 3 encoder.

**Selecting the best explanation per feature.** Inspired by Choi et al. (2024), who use a fine-tuned scorer to identify the best explanations out of a set of candidates, we adopt a rank-based voting strategy to select the top explanation across the three explanation methods for each SAE feature. Specifically, each evaluation metric independently ranks each explanation method. Then, the explanation with the lowest (best) total rank is selected. In case of a tie, the explanation is chosen at random.

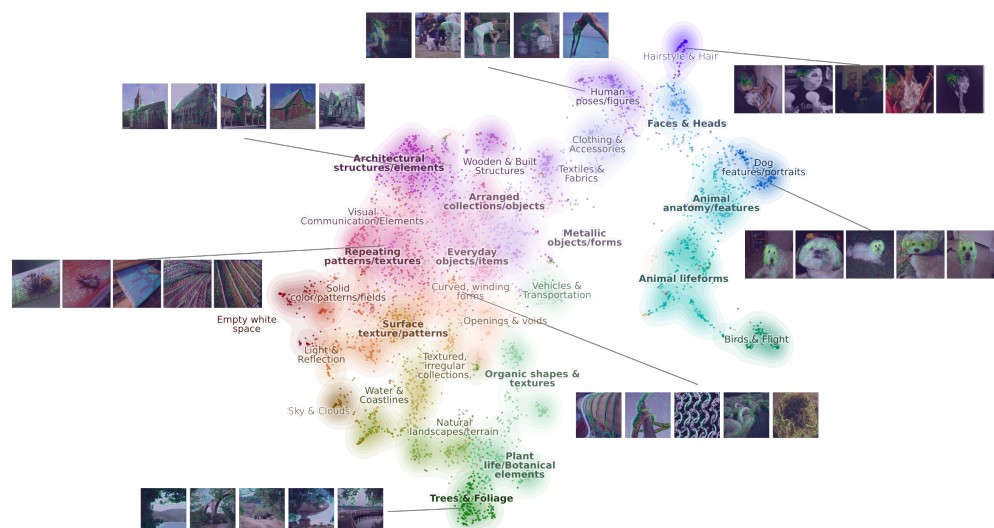

Figure 6: UMAP visualization of SAE feature explanations, Gemma 3 vision encoder, middle layer.

To ensure we select interpretable features with meaningful explanations, we discard features whose selected explanation has an IoU score (Equation (8)) below 0.2 or synthetic activation score (Equation (10)) below 0.3. This filtering step leaves around 5,000 out of the original 7,690 alive features with assigned explanation. As shown in Figure 8, *Steering-informed Top-k* is selected more frequently. Notably, *Steering* and *Top-k* explanations are selected at similar rates.

**Visualizing the SAE feature space.** After selecting high-quality explanations, we compute semantic embeddings using a sentence similarity model[8] (Reimers and Gurevych, 2019). The projected 2D UMAP (McInnes et al., 2018) representation of these embeddings is shown in Figure 6, where the clusters are obtained via k-means algorithm (Lloyd, 1982) with $k = 30$. To facilitate interpretation, we assign a label to each cluster by giving Gemma 3 27B a random sample of 20 explanations from that cluster.

Since the SAE is trained on ImageNet, the learned features seem to capture concepts prevalent in the dataset, such as humans (*Human poses/figures*), animals (*Animal lifeforms*), and natural scenes (*Trees & Foliage*). While many explanations correspond to high-level semantic categories (*e.g., Vehicles, Clothing*), which aligns with expectations for middle-layer features (Cammarata et al., 2020), we also observe features at lower levels of abstraction. These include perceptual features like *Repeating patterns/textures* and *Surface texture/patterns*. Notably, as shown in Figure 8, *Steering* allows obtaining these low-level features.

**Finding features previously thought unique to DinoV2.** The semantic space of explanation embeddings enables targeted retrieval of features aligned with user-specified concepts. As a proof of concept, we search for features previously identified by Thasarathan et al. (2025) as unique to DinoV2 (Oquab et al., 2024), a vision model trained without language supervision. Contrary to prior claims, we found features seemingly representing *depth* (Figure 7 top) and *perspective* (Figure 7 bottom) in our SigLIP SAE. For instance, the *depth* feature is described by the *Steering* explanation as: "*Blurred, out-of-focus background creating a sense of depth and indistinctness.*", and the *perspective* feature as "*Long, receding perspective created by converging lines, evoking a sense of depth and distance*". While anecdotal, these findings demonstrate the utility of combining steering-based explanations with semantic search to uncover conceptual overlap across models.

---

[8]We use `sentence-transformers/all-mpnet-base-v2`.

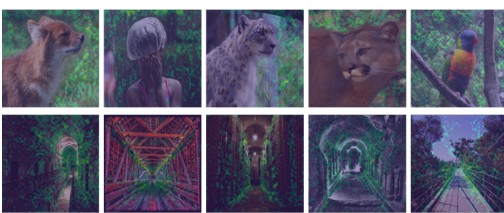
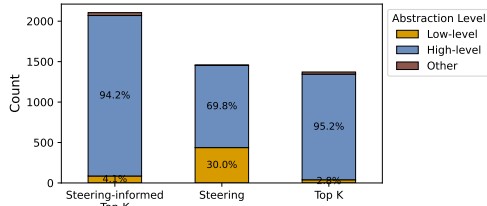

Figure 7: *Depth* (top) and *perspective* (bottom) features previously found as unique to Dinov2, surfaced via steering explanations.

Figure 8: Count of selected explanations by each method. Each bar shows the level of abstraction of the selected explanations.

## 6 RELATED WORK

Interpretability in vision models has seen rapid progress (Joseph, 2024), with recent work aiming at mapping internal representations to natural language. A key strategy has been to leverage CLIP's shared image-text embedding space to align vision model features with human-understandable concepts (Gandelsman et al., 2023; Bhalla et al., 2024).

In parallel, mechanistic interpretability has advanced our understanding of LLMs (Ferrando et al., 2024), with SAEs revealing interpretable features (Bricken et al., 2023). Recently, SAEs have been applied to vision models (Fry, 2024; Lim et al., 2025; Thasarathan et al., 2025; Rajaram et al., 2025; Venhoff et al., 2025; Shabalin et al., 2025), revealing semantically meaningful features. Yet, interpreting thousands of features remains a bottleneck, highlighting the need for automated solutions.

Automated interpretability in LLMs has traditionally followed 'input-centric' strategies, where explanations are generated from top-activating inputs (Bills et al., 2023; Choi et al., 2024). This input-centric method perspective has been extended to vision SAEs (Zhang et al., 2024; Xu et al., 2025; Rao et al., 2024), where top-activating images are used instead. To address input-centric limitations, recent work has shifted toward output-centric explanations. Gur-Arieh et al. (2025) propose *VocabProj* and *TokenChange* to reveal which outputs are causally tied to specific features. Similarly, Paulo et al. (2024) introduce an intervention-based metric to assess explanation quality through causal influence. In vision models, output-centric causal approaches based on steering have also emerged, though applications have so far remained confined to within-model interventions (Joseph et al., 2025; Lim et al., 2025; Stevens et al., 2025), while we propose leveraging a language model to generate the explanation on the intervened vision encoder.

Closely related to our work are efforts on self-explaining features in LLMs. *Patchscopes* (Ghandeharioun et al., 2024; Chen et al., 2024) uses activation patching to transfer representations and generate causal explanations. Kharlapenko et al. (2024) extend this idea to SAEs, enabling the model to act as its own explainer by describing its features. Our work builds on these trends by proposing a causal, output-centric method for interpreting SAE features in vision models through direct intervention and language-based explanation.

## 7 CONCLUSIONS

This work presents a new framework for automatically interpreting features in vision models. By steering the encoder with targeted feature interventions alone, and leveraging a language model as the explainer, we generate feature explanations in an efficient and scalable way. While *Steering* overall tends to underperform *Top-k* method, it avoids their contextual biases and is particularly effective at surfacing lower-level features. Moreover, combining both approaches enables the identification of higher-quality explanations, highlighting their complementary nature. Explanation quality also scales consistently with language model size, suggesting that as LMs continue to advance, steering-based explanations will become increasingly informative and precise. The hybrid *Steering-informed Top-k* approach consistently produces the highest-quality explanations across evaluation metrics, demonstrating the value of integrating causal interventions with input-based methods.

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

## A    SAE TRAINING DETAILS

For training the SAEs, we used the `dictionary_learning` library (Marks et al., 2024). All SAEs were optimized using the Adam optimizer with a learning rate of $3 \times 10^{-4}$, $\beta_1 = 0.9$, and $\beta_2 = 0.99$. Training was conducted over a single epoch of the ImageNet training set (1.28M images) with a batch size of 8192. We enforced a sparsity constraint of 25 active features per patch position.

Model activations from `HuggingFace` (Wolf et al., 2020) were cached on-the-fly during training. We maintained a buffer of 500 million activations, from which we randomly sampled. When the buffer was depleted to half capacity, it was refilled with new activations.

## B    STATISTICAL TEST DETAILS

To assess statistical significance across explanation methods, we conduct pairwise one-tailed tests for each evaluation metric and masking type. Since evaluation scores are not normally distributed, as verified via a Shapiro-Wilk test, we apply the nonparametric Mann-Whitney U test. An explanation method is considered statistically significant if it is stochastically greater than both alternatives (with $p < 0.05$).

## C    LATER LAYER RESULTS

Table 3: Explanation evaluation metrics for the later layer SAE of Gemma 3 vision encoder. Except for AUROC, mean scores are reported, and statistical significance is assessed pairwise between methods. A value is underlined if it is significantly higher (with $p < 0.05$) than both other methods in the same column.

| Explanation Method | IoU Score | | AUROC | | Synth. Act. Score | | CLIP Score | |
|---|---|---|---|---|---|---|---|---|
| | Masks | Heatmaps | Masks | Heatmaps | Masks | Heatmaps | Masks | Heatmaps |
| Steering | 0.204 | | 0.773 | | 1.473 | | 0.182 | |
| Top-k | 0.194 | 0.186 | 0.782 | 0.857 | 1.453 | 1.609 | 0.188 | 0.187 |
| Steering-informed Top-k | 0.196 | 0.183 | 0.810 | 0.908 | 1.691 | 2.156 | 0.190 | 0.186 |

## D    *Top-k* EXPLANATION EVALUATION SCORES AS A FUNCTION OF SEMANTIC SIMILARITY BETWEEN *Steering* AND *Top-k* EXPLANATIONS

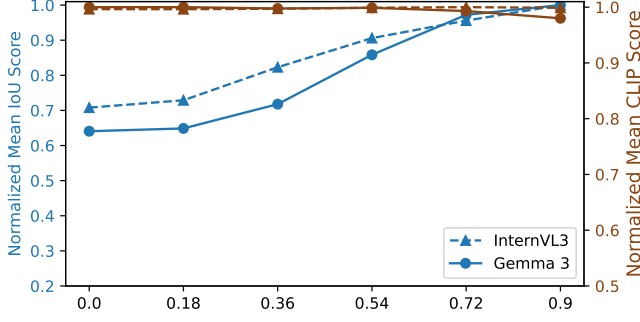

Figure 9: IoU Score and CLIP score values for *Top-k* method as a function of the similarity with *Steering* explanations.

## E   GEMMA 3 IOU AND CLIP SCORES OF *Steering* METHOD AS A FUNCTION OF THE SIZE OF THE LM M$_{\text{SUBJ}}$

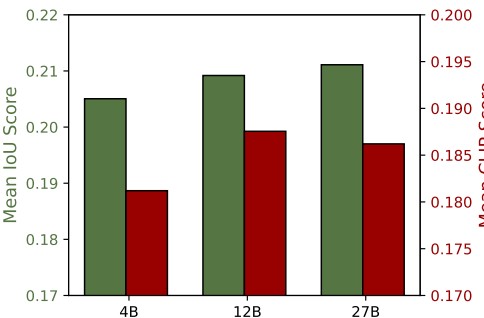

Figure 10: IoU Score and CLIP score values as a function of dataset size, for Masks Top-k method.

## F   FLOPS ESTIMATION

We compare the approximate Floating Point Operations (FLOPs) for generating explanations, using the estimate $2 \times$ Parameters $\times$ Tokens for a model forward pass (Kaplan et al., 2020). Let:

- $N_{\text{eval}} = |\mathcal{D}^{\text{eval}}|$: size of the evaluation image set.

- $P_{\text{sub}}$: parameters of the subject model $m_{\text{sub}}$ (also serving as $m_{\text{exp}}$).

- $P_{\text{SAE\_enc}} = d_{\text{model}} \cdot d_{\text{SAE}}$: parameters for the SAE.

- $T_{\text{img}}$: per image token representations (for $m_{\text{sub}}$ input, for SAE processing per image, and for the empty image $\tilde{I}$. E.g., 4096 for Gemma 3).

- $T_{\text{prompt}}$: token count for the textual prompt.

- $T_{\text{expl}}$: max tokens in the explanation.

- $k$: number of top images selected.

***Top-k* Explanations.**    This method consists of two main computational stages:

1. **Dataset Precomputation** (typically a one-time process to identify top-$k$ activating images for features): It involves processing all $N_{\text{eval}}$ images through $m_{\text{sub}}$, followed by SAE encoding for each representation using $\boldsymbol{W}_{\text{enc}}$. FLOPs$_{\text{precompute}} \approx N_{\text{eval}} \cdot T_{\text{img}} \cdot 2 \cdot (P_{\text{sub}} + P_{\text{SAE\_enc}})$. Aggregation and sorting costs are generally minor in comparison.

2. **Per-feature Explanation Generation**: The explainer model $m_{\text{sub}}$ is conditioned on the prompt and the $k$ selected images. FLOPs$_{\text{gen}} \approx 2 \cdot P_{\text{sub}} \cdot (T_{\text{prompt}} + k \cdot T_{\text{img}} + T_{\text{expl}})$.

The total cost is dominated by FLOPs$_{\text{precompute}}$ when $N_{\text{eval}}$ is large.

***Steering*-based Explanations.**    This approach avoids the dataset precomputation. An explanation for each feature $i$ is generated via a single forward pass of $m_{\text{sub}}$ from an intervention using the pre-defined SAE feature direction $\boldsymbol{W}_{\text{dec}}[i, :]$:

- **Per-feature Explanation Generation**: FLOPs$_{\text{steer}} \approx 2 \cdot P_{\text{sub}} \cdot (T_{\text{prompt}} + T_{\text{img}} + T_{\text{expl}})$. The costs for retrieving the SAE feature direction and applying the intervention (vector operations) are also incurred, in addition to the forward pass captured by the formula above.

***Steering*-informed *Top-k* Explanations.**    This method combines the dataset precomputation with an intervened generation step:

1. **Dataset Precomputation**: This stage is identical to the corresponding stage in the *Top-k* method, incurring FLOPs$_{\text{precompute}}$ as defined above.

2. **Per-feature Explanation Generation**: Similar to standard *Top-k* generation, but with an intervention. The computational cost for generation remains approximated by $\text{FLOPs}_{\text{gen}}$ as defined for *Top-k* explanations. The costs for retrieving and applying the intervention are also incurred here, similar to the pure *Steering-based* method. This method achieves the best results at a comparable cost.

## G    PROMPTS

---

**Steering Prompt**

```
You are given an image highlighting a visual or semantic element. This element may
range from a low-level visual feature to a high-level abstract concept. Your task is to
describe this element in a single, clear sentence. If the element is a high-level
abstract concept, describe it as such; otherwise, describe its visual patterns.
Favor a more general interpretation. Start the highlighted element description
with \"The highlighted element in the image is a\".
```

Figure 11: Prompt used for obtaining explanations for the *Steering* method.

---

**Top-k and Steering-informed Prompt (Masks)**

```
You are given set of images highlighting a visual or semantic element. The patches of
the images not showing the element are masked out, giving the impression of a
pixelated image. This element may range from a low-level visual feature to a high-level
abstract concept. Your task is to describe this element in a single, clear sentence.
If the element is a high-level abstract concept, describe it as such; otherwise,
describe its visual patterns. Favor a more general interpretation. Provide a single
description for the highlighted element appearing in all images, and please ignore the
pixelated effect of the mask when describing the element. Start the highlighted element
description with \"The highlighted element in the image is a\".
```

Figure 12: Prompt used for obtaining explanations for the *Top-k* and *Steering-informed Top-k* method with Masks.

---

**Top-k and Steering-informed Prompt (Heatmaps)**

```
You are given set of images highlighting a visual or semantic element. The patches of
the images showing the element are highlighted with a green heatmap. This element may
range from a low-level visual feature to a high-level abstract concept. Your task is
to describe this element in a single, clear sentence. If the element is a high-level
abstract concept, describe it as such; otherwise, describe its visual patterns. Favor
a more general interpretation. Provide a single description for the highlighted element
appearing in all images, and please ignore the overlayed green heatmap when describing
the element. Start the highlighted element description with \"The highlighted element
in the image is a\".
```

Figure 13: Prompt used for obtaining explanations for the *Top-k* and *Steering-informed Top-k* method with Heatmaps.

## H    DEMO

The demo interface is designed to visualize and compare the different types of explanations computed in our analysis. On the left panel, users can select the different layers, switch between models, and search for explanation examples that contain specific keywords.

On the right, the main panel includes three view options. In the *Feature Details* view, the top section displays the explanations generated by the three methods discussed in the paper: *Top-k*, *Steering*, and *Steering-informed Top-k* (referred to as Top-k w/ Steering). Each explanation is shown using both masks and heatmaps (the latter are labeled with *Heatmap* in the name). In the bottom section of this view, the top activating images for each feature feature are displayed.

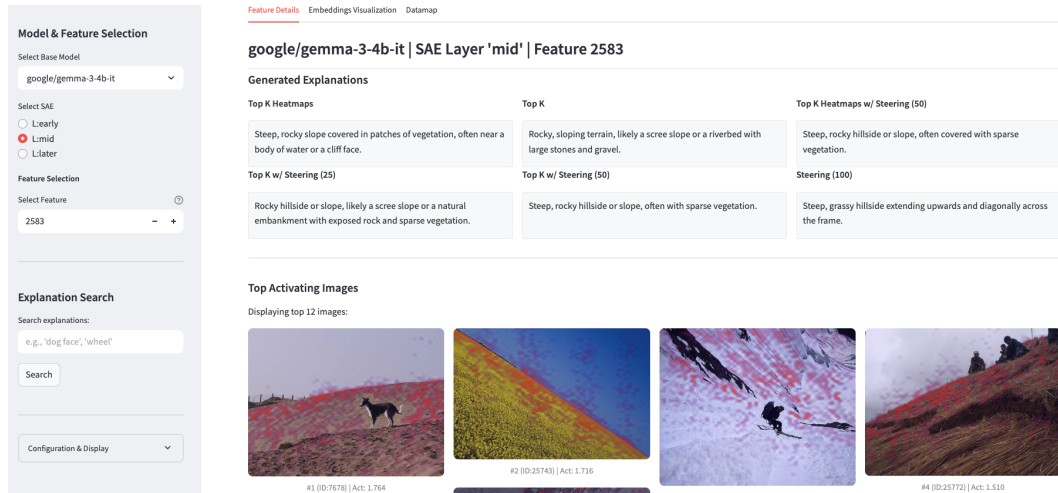

Figure 14: *Feature Details* view of the demo interface. The top section shows explanations for a selected feature using different methods (*Top-k*, *Steering*, and *Steering-informed Top-k*), using both masks and heatmaps. The bottom section displays the top activating images for the selected feature.

The other two views, *Embeddings Visualization* and *Datamap*, show the UMAP projection of all explanations. In the *Datamap* view, the clusters are shown with their corresponding topics.

## I USE OF EXISTING ASSETS

We use the following assets in our work:

### MODELS

Table 4: The list of models used in this work.

| Model | Link | License |
|-------|------|---------|
| Gemma 3 (Team et al., 2025) | Hugging Face (Google) | Gemma Terms of Use [9] |
| InternVL3-14B (Zhu et al., 2025) | Hugging Face (OpenGVLab) | Apache 2.0 |
| CLIP (Radford et al., 2021) | Hugging Face (OpenAI) | MIT License |
| SAM2 (Ravi et al., 2025b) | Hugging Face (Meta) | Apache 2.0 |
| Stable Diffusion (Esser et al., 2024) | Hugging Face (Stability AI) | CreativeML OpenRAIL M license |
| all-mpnet-base-v2 (Reimers and Gurevych, 2019) | HuggingFace | Apache 2.0 |

### DATASETS

Table 5: The list of datasets used in this work.

| Dataset | Link | License |
|---------|------|---------|
| ImageNet (Deng et al., 2009) | Official Website | Custom (Non-commercial) |

## J COMPUTE RESOURCES

All training and evaluation experiments were run on a single node of 4x NVIDIA Hopper H100 64GB GPUs. The demo website runs on a machine with 2x NVIDIA 4090 GPUs. Each Gemma 3 SAE training took approximately 6 hours on 1 GPU, and 3 hours for InternVL3.

