# OpenReview forum: "Language Models Can Explain Visual Features via Causal Interventions"
_ICLR.cc/2026/Conference — ICLR 2026 Conference Withdrawn Submission_

### Official Review · Reviewer_3AQY · 2025-10-19

**Soundness:** 2
**Presentation:** 3
**Contribution:** 2
**Rating:** 4
**Confidence:** 4

**Summary:**

The paper aims to explain and understand the visual concepts encoded in the vision encoder of MLLMs. This is achieved by first mapping the embeddings to a higher dimension using a trained Sparse Autoencoder (SAE). By steering the vision encoder’s residual stream with individual SAE features—while feeding it an empty image—the MLLM is prompted to describe the visual concept represented by that feature. Experiments on Gemma 3 and Intern VL3 vision encoders demonstrate that Steering offers a scalable alternative that complements traditional approaches based on input examples

**Strengths:**

1. Important research question: The paper addresses a significant research problem: understanding the visual concepts encoded in the visual encoder of MLLMs.
2. Rich visualizations and discussions: The paper provides many meaningful and interesting visualizations, along with detailed discussions of the results in Section 5.

**Weaknesses:**

1. Lack of clarity and theoretical grounding for the method.

    The description of the method raises several questions and uncertainties:

    1a. What is meant by “an empty image”? In theory, it should be an image that contains no meaningful information for the visual encoder. However, it is unclear what type of image is used in practice and how it is ensured that the image carries no information to the visual encoder.

    1b. The practical implementation of the “do-operator” (the steering operation) is unclear. How exactly is this operation performed?

    1c. Beyond the clarity of the “do-operator,” there is no theoretical justification for why adding the do-operator in Equation (6) would lead to better explanations of visual concepts compared to simply fitting the top-k images.

2. Lack of clarity in implementation:

    2a. Why is the SAE trained on ImageNet? How does the choice of the dataset used for training impact the explanations?

    2b. Which layer of the vision encoder is used for the SAE? How does the choice of layer affect the explanations? Would using early or later layers yield different visual concepts?

    2c. In the simulation-based evaluation, "a segmentation model (e.g., SAM2) is used to generate binary masks corresponding to the concepts described in the explanations". However, segmentation models like SAM2 cannot generate segmentations based on text prompts. What specific segmentation model is used in the experiments, and does the model give precise masks corresponding to the text explanations?

3. Limited technical contribution: Steering alone does not yield better explanation performance than the top-k approach. While combining Steering with top-k improves performance, the technical contribution of this improvement seems limited. Moreover, there is no explanation or theoretical support for why adding Steering on top of top-k leads to better explanations.

**Questions:**

See weaknesses above.

---

### Official Review · Reviewer_BGAz · 2025-10-29

**Soundness:** 2
**Presentation:** 2
**Contribution:** 2
**Rating:** 2
**Confidence:** 4

**Summary:**

This paper studies automated interpretability for vision models by using causal interventions (“steering”) of Sparse Autoencoder (SAE) features into a vision encoder and asking a vision–language model to describe what it “sees.”

**Strengths:**

The topic is timely and broadly useful: automating explanations for thousands of learned visual features addresses a real bottleneck in interpretability research.

The steering / intervention idea is conceptually appealing: generating explanations from an intervention on an empty image is efficient and offers a causal perspective that can mitigate some contextual biases of top-k methods.

**Weaknesses:**

Paper organization & missing SOTA context. Related work and state-of-the-art baselines are not placed up front; as written the reader must wait too long to understand the landscape and how the contribution differs. This makes the method and its novelty hard to evaluate.

Incomplete baseline comparisons. Experiments do not clearly include or explain comparisons to the most relevant recent automated explainers (e.g., VocabProj/TokenChange, Patchscopes, prior Top-k pipelines, and other causal/steering methods). It is unclear whether gains come from the idea itself or from evaluation choices.

Marginal technical novelty. The core technical move (add SAE dictionary vector into residual stream, prompt LLM on blank/top-k images) is elegant but incremental relative to prior intervention/patching and output-centric work; the manuscript does not present a novel algorithmic contribution or theoretical insight that goes beyond combining existing pieces.

Figure clarity and exposition. Figures (notably Figure 1) are hard to parse: they do not clearly distinguish “current/problem” vs “proposed” pipeline, lack notation, and are not self-contained. Several figures are not referenced or explained in the flow.

Intervention formalization and sensitivity. The intervention is described verbally but lacks precise math and ablations: how is α chosen, how robust are explanations to α, is the intervention linear add or something else, and are there layer/position sensitivity analyses?

**Questions:**

For readability and evaluation: will you move the Related Work immediately after the Introduction and explicitly map prior methods (Top-k, Patchscopes, VocabProj/TokenChange, prior steering/patching) to the exact differences of your approach? Can you add a small table that lists what each prior method requires (eval images, segmentation models, forward passes, human labels) so readers can see the gap you fill?

Which specific SOTA baselines did you compare to, and can you add direct quantitative baselines for Patchscopes, VocabProj/TokenChange, and any recent Top-k variants? Please include details so we can see whether Steering itself, or the hybrid selection rule, provides the gain.

What is the algorithmic novelty beyond composing SAE steering + VLM explanation? Can you (a) identify a clear technical contribution (e.g., theory/guarantee about when steering yields faithful explanations, or a novel intervention protocol), or (b) otherwise tone down novelty claims and emphasize empirical utility?

Please revise Figure 1 (and any pipeline figures) so each subpanel is self-contained: explicitly mark (A) baseline / problem, (B) proposed steering operation (with equations and shapes), and (C) your hybrid pipeline. Also ensure each figure has a caption that a reader can understand without reading the main text.

How sensitive are results to choices of k (top-k), evaluation dataset size, and to the segmentation model (SAM2) used for IoU? Please add sensitivity analyses (vary k, vary dataset size, show a small table comparing IoU when using different mseg).

---

### Official Review · Reviewer_nJLo · 2025-10-31

**Soundness:** 2
**Presentation:** 2
**Contribution:** 2
**Rating:** 4
**Confidence:** 4

**Summary:**

This work aims to propose an automatic approach to evaluate the interpretability of concepts discovered by sparse autoencoders in vision models. Concretely, this work obtains the language based interpretation via feeding an empty image into the vision encoder and steering the SAE features in a vision language model. Experimental results show how the approach reduces contextual bias compared to example based interpretation, and shows how it achieves better explanation metrics compared to pure steering based methods.

**Strengths:**

- This method proposes an approach to automatically interpret the meaning of SAE features, which is an important aspect in existing SAE based interpretation research.
- Considering a blank image in generating the explanation in vision language models helps to reduce the contextual bias.
- Given the difficulty of evaluating pure text based explanations, several metrics are adopted to justify the benefits of the proposed approach.

**Weaknesses:**

-  Steering is not new in interpretation and is a standard operation in causality:
Directly steering SAE features to understand its meaning is already adopted and widely studied in LLM [1]. The paper does not discuss the difference between doing so in LLM and VLM to justify its contribution.
[1] https://docs.neuronpedia.org/steering

-	It seems the major novelty from the method perspective comes from explaining an empty image compared to prior approaches. But the motivation of doing so is not clearly explained: why not a random image / a gaussian noise / a black image? Why still using the embedding of a blank image instead of directly using 0 embedding?
-	[line 66] “self-explaining” does not follow prior works’ common usage of this word [2] and authors do not clearly define its meaning in its context neither.
[2] Towards Robust Interpretability with Self-Explaining Neural Networks. NIPS2018

-	It’s not very convincing to me why conditioning on top activating images while steering SAE features in a vision-language model to output some texts can be considered as “interpreting SAE features”. Any change in the input image space could be also considered as an intervention, and the generated explanation is also influenced by any change in that space. SAE features have some influence on the generated textual explanation, but the textual explanation is not necessarily explaining the SAE feature. I am wondering whether fixing the same set of conditioning images while changing the SAE feature would simply yield similar explanation. To which extend is the generated explanation influenced by the topk images and to which extend is it influenced by the SAE feature?
-	[line 71-72] The key motivation of why the proposed method is important/complementary is described in a very vague way: “overcoming some of the explanation biases these methods introduce, while surfacing lower-level features.”
-	[line 76] The authors claim to “combine the best of both approaches”. But what is good in existing example based interpretation is not well explained. Ideally, why existing advantages are further maintained in the new approach should also be explained to support the claim “combine the best of both approaches”.
-	[line 286] compared to what approach is the proposed approach more scalable? Example based interpretation is actually also quite scalable without much human effort.

**Questions:**

[line 14] What’s the problem of existing example based interpretation methods?

[line 12] “Explaining these features without requiring human intervention remains an open challenge”. But prior works do not require much human intervention neither?

[line 19] What is complemented? Is the existing example based approach not scalable?

Are you using the SAE reconstructed feature during explaining the empty image or the original feature in that layer?

---

### Note · Authors · 2025-11-14

**Comment:**

We thank the reviewers for the time they dedicated to evaluating our paper. We have decided to withdraw the submission, but we will incorporate the feedback into future revised versions. We will address specific concerns in the following comment.

**Withdrawal Confirmation:**

I have read and agree with the venue's withdrawal policy on behalf of myself and my co-authors.